# Characterization and management of human-wildlife conflicts in mid-hills outside protected areas of Gandaki province, Nepal

Kedar Baral[1,2]*, Hari P. Sharma[3], Bhagawat Rimal[4], Khum Thapa-Magar[5], Rameshwar Bhattarai[6], Ripu M. Kunwar[7]*, Achyut Aryal[1], Weihong Ji[1]

1 School of Natural and Computational Sciences, Massey University, Palmerston North, New Zealand, 2 Division Forest Office, Kaski, Department of Forest and Soil Conservation, Kathmandu, Nepal, 3 Central Department of Zoology, Institute of Science and Technology, Tribhuvan University, Kathmandu, Nepal, 4 College of Applied Science (CAS-Nepal), Tribhuvan University, Kathmandu, Nepal, 5 Forest and Rangeland Stewardship Department, Colorado State University, Fort Collins, CO, United States of America, 6 Asian Center for Development, Kathmandu, Nepal, 7 Department of Geosciences, Florida Atlantic University, Boca Raton, FL, United States of America

* kbaral@massey.ac.nz (KB); rkunwar@fau.edu (RMK)

**Data Availability Statement:** All relevant data are within the paper and its Supporting Information files.

## Abstract

With the intent to better management human wildlife conflict (HWC) and wildlife conservation in mid-hills outside protected areas of Gandaki province, Nepal, we analyzed the patterns and drivers of HWC. Using data collected from literature, government records and questionnaire survey, we investigated temporal, seasonal and spatial distribution of human casualties caused by wildlife attacks. We also appraised the perception of local people towards wildlife conservation. We have recorded 77 cases (69 human injuries and 8 mortalities) during the period of nine year between 2011 and 2019. The number of wildlife attacks increased over this period. Wildlife attacks were more frequent in winter with 50% (42) of attacks occurred between September and December. Common leopard (*Panthera pardus*) and Himalayan black bear (*Ursus thibetanus laniger*) were the major species involved in these conflicts. Common leopard was the most feared species that causes highest number of human mortalities (87%, n = 67); the most severe type of HWC outcome. Forty-eight percent (n = 37) attacks were reported at human settlement areas followed by 27% attacks in agriculture land (n = 21) and 24% (n = 19) in forest. Generalized linear model analysis on spatial variables showed that the probability of human attacks increases with decreasing elevation (β = -0.0021, Z = -1.762, *p* = 0.078) and distance from the forest (β = -0.608, Z = -0.789, *p* = 0.429). We recommend to decrease habitat degradation / fragmentation, carry out habitat management program within forest to increase prey availability to decrease the wildlife invasion into human settlement area, and decrease dependency of people on forest resources by providing alternative livelihood opportunities. Simplified relief fund distribution mechanism at local level also helps alleviate the impact of HWC. The knowledge obtained by this study and management measures are important for better human-wildlife co-existence.

**Funding:** The funders had no role in study design, data collection and analysis, decision to publish, or preparation of the manuscript.

## Introduction

The Human-Wildlife Conflict (HWC) refers to the interactions between human and wild animals that results in negative consequences on livelihood and life of people and or wild animals [1]. HWC dates back to human prehistory. The earliest forms of HWC occurred in the form of predation of early hominoids by wild animals or vice-versa [2]. HWC occurs in different contexts involving a range of animal taxonomic groups [3–5]. It is a common issue in the Himalaya region where wildlife and people co-exist [6] and share the limited resources [5]. This region exhibit great propensity for HWC due to its rich biodiversity, heavy reliance of people on forests, cropland and animal husbandry for livelihoods [7, 8]. A high degree of dependency on forest ecosystems and prevalent poverty has led to unsustainable extraction of forest products and conversion of forests into agricultural land [9]. HWC incidences have increased as poaching, deforestation, habitat degradation and fragmentation and overexploitation are escalated [10] with increase in human population [11, 12].

The common forms of HWC in the Nepal Himalaya are crop raiding, property damage, livestock depredation and human-injuries and mortalities caused by wildlife attack [13, 14]. Among these types of HWCs, the latter two have the highest cost [15, 16]. Common leopard (*Panthera pardus*) causes the highest number of human attacks in Nepal, followed by wild elephants [6]. The occurrence of leopard attacking humans in Nepal is higher than that of anywhere else within leopard distribution range [17]. Leopard attacks occurred in all regions outside the protected area system (PAs) and mid-hills of Nepal [18, 19] which have severely impacted the traditional agricultural and farming practices [20–22].

Most of the protected areas (PAs) in Nepal are established to conserve large mammals. HWC caused by the large mammals such as Asian elephants (*Elephas maximus*), greater one-horned rhinoceros (*Rhinoceros unicornis*) and tigers (*P. tigris*) has been increasing due to increase in their populations since the establishment and implementation of PAs [23, 24]. The increased competition among animals within PAs may have triggered the invasion of wildlife to areas outside PAs, resulting in frequent HWCs [7]. In the last five years, over two third of HWC incidences in Nepal occurred outside of PAs [25, 26]. Around 40% of the forest areas outside PAs have been managed under the community forestry programme [6]. Due to the successful implementation of community-based forestry program in the last three decades, degraded forests have been restored and recovered the wildlife population in mid-hills outside PAs, promoting free movement of wildlife [27–30]. Such situation has increased the encounter of local people with wildlife resulting in increased HWC.

The forest habitats including community forests outside PAs in mid-hills ecosystem of Nepal are surrounded by the large human population [21], and hence the landscapes are human dominated. HWC is mostly associated with the local livelihood [31, 32] since the dependency of people is very high on forest resources to sustain their life. The use of forest resources / area for summer grazing, herding, collection of plants and harvesting of forest products i.e. timber, fuel-wood from forests and forest fringes are still persistent among people of studied districts [33, 34]. HWCs occurred in human settlement outside PAs while people carrying out such activities [35–37]. However, the management of HWC at landscape level addressing the issues outside and inside PA are limited [38]. Hence, management of HWC outside PAs and decreasing dependency of people on forests resources are therefore urged [32].

The studies on the causes, and characteristics of HWC outside the PAs in mid-hills and central Nepal are scarce [15, 26]. Outside PAs of Nepal, HWC incidences are very frequent [37] and Himalayan black bear and leopard are the main wildlife species responsible for attacking human [39, 40]. Their ecology and interactions with human are poorly studied. Understanding

the patterns and causes of Human-Wildlife interactions outside PAs of Nepal is most pertinent in formulating conservation policies to mitigate HWC and improving the local livelihood. In this study, we examine the extent and magnitude of HWC associated with casualties and injuries of wildlife and human, assess the spatio-temporal distribution of HWCs involving human casualties and recommend the strategies for future conservation planning.

## Methods

### Study area and site description

Five districts, Parbat, Lamjung, Syangja, Tanahun and Kaski (hereafter referred to as PLSTK) (27˚55' - 28˚26' N, 83˚58'– 84˚25' E) were selected as study area (Fig 1). These districts have human population of about 1.4 million, many of whom live in remote and isolated areas with poor access to markets and easy access to forest resources. These five districts cover the area of approximately 7,000 km$^2$ and represent lowlands (60 to 1000 m asl), mid-hills (1000 to 3000 m asl) and high mountainous area (3000 m to 5000 m asl) of central Nepal. The high mountainous part of Kaski and Lamjung districts shares 11% of Annapurna Conservation Area (7,629 km$^2$), the largest PA of Nepal. The mid-hill region (also known as middle Mountain) is characterized by sub-tropical and temperate bio-climates with a great variety of terrain types, ecosystems and wildlife [41, 42]. Broad-leaved, coniferous and oak-laurel forests are common in mid-hills [43]. Over 50% of country's' population lives in hills and mountains [44]. The hills and mountains not only have the highest percentage of poverty (42%) on average, but it is also increasing [45].

Within the study sites, Devghat, Tanahun district represents the lowest elevation and tropical bio-climate (187 m asl) whereas the Lwang-Ghalel (~ 2500 m) of Kaski district, Chimkeswari (2325 m) of Tanahun, Panchmool (~ 2600 m) of Syangja district, Dahare-Deurali (~ 2600 m) of Parbat district and Bahundanda (~ 2600 m) of Lamjung district represent the highest elevations and temperate bio-climates. The study sites cover a big section of Chitwan Annapurna Landscape (CHAL) corridor. The CHAL represents north-south corridor and connects alpine high mountains in north to tropical lowlands in south through Gandaki watershed. The Gandaki watershed connects Marsyangdi (Lamjung), Madi (Tanahun), Setigandaki (Kaski) and Kaligandaki (Parbat and Syangja) sub-river basins. These sub-river basins boast high biodiversity and supports rich natural, social and cultural heritage [43]. It is an important transit route for migratory birds and is home to endangered species such as common leopard, red panda (*Ailurus fulgens*) and Himalayan black bear [30, 46].

Total population of PLSTK districts is 1318848 [47], with population density 219 people per KM$^2$. Majority of the people (76.86%) live in the rural area and agriculture is the mainstay of local livelihood. Most of the rural settlements are close to the forest area and fuelwood is the major source of energy [48]. Other livelihood strategies include animal husbandry, summer grazing, and collection, use, and trade of forest products. The dominant ethnic groups Gurung and Magar comprised of about 35% of the total population of the area [47], are mountain dwellers, sheep herders, cattle grazers and trans-himalayan traders [34] who often confront with wildlife in the mountain rangelands [33, 49, 50].

### Data collection

We received human ethics research approval 4000023041 from Massey University New Zealand for our research. Prior to fieldwork and interview, we also received written consents from the division forest offices and oral consents from participating individuals. HWC data of the study districts were collected from both published articles and the government reports relevant to this study. HWC databases of PLSTK division forest offices between 2011 and 2019 were

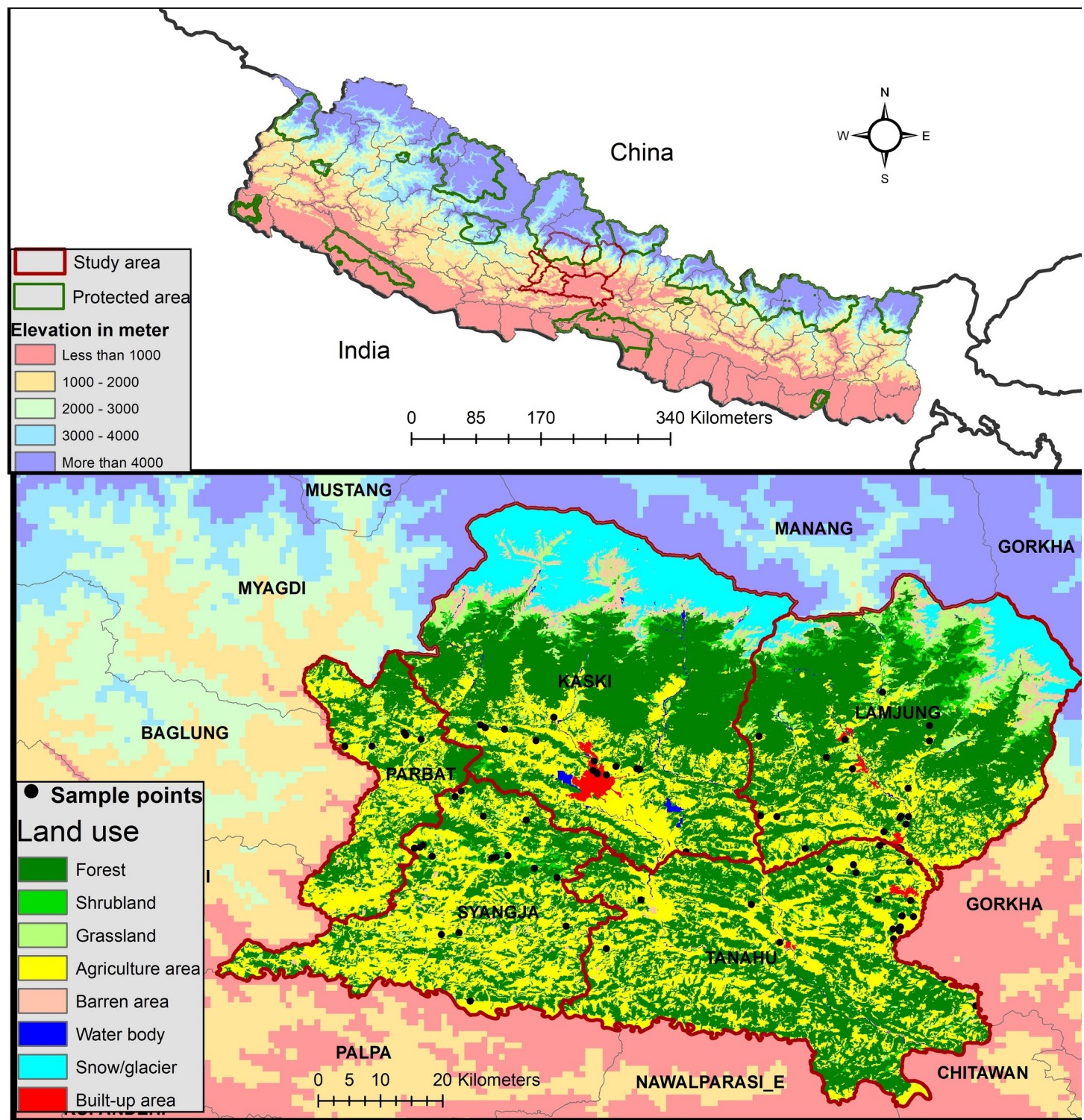

**Fig 1. Location of study area showing the districts, elevation, land use and points of wildlife attacks.** "Republished from [51] under a CC BY license, with permission from ICIMOD, original copyright 2010".

reviewed. Data records only after 2010 were logged when the official recording of wildlife damage for compensation scheme was formally started. Information of 77 cases of human attacks (injury and death) was collected from the records of PLSTK division forest offices (DFO). Divisional forest officers were consulted to crosscheck the data and gather S1 File. Field visit was made to meet the victims and persons accompanying the victim, and to observe the sites of attack. We visited the sites of 12 attacks including all eight human deaths. Number of the field visit on the PLSTK districts was proportionately selected based on the total number of the conflict events in the districts. Since Tanahun district has high incidents of human-wildlife conflict, we visited all six sites (Nareshowar tar, Purkot, Khahare, Bhansar, Samjur and Mirlung) of human death in Tanahun. Field observations of conflict sites and checklist surveys of the victims and accompanying people were carried out between March and December 2019 to verify the sites and conflicts, and to collect additional information regarding the perception of people and management options. We conducted 36 interviews with the victims of HWCs and another 21 interviews with the persons accompanying the victim during the attack, or immediate family member who was well aware of attack details. Personal details during the interview were maintained confidential and in the analysis process the name were omitted by recording a code to keep the interviewees anonymous. We used a semi-structured checklist to conduct the interviews. Interviews were conducted in Nepali language with the help of a forest ranger and local assistants.

## Data analysis

We collected four types of variables associated with human-wildlife attacks; species and number of wildlife involved in human attacks, temporal variables include year, month, season and time of attacks (0–4, 5–8, 9–12, 13–16, 17–20 and 21–24 hours), socio-demographic variables include the gender, occupation, and age group of the victims, and spatial variables include district, elevation, distance between point of attack and forest, and land use (forest, agricultural land, road and settlement) (S1 File). We classified attack locations according to land use as forests, agricultural fields, and settlement area. The distance from the casualties and nearest forest edge was measured in Google Earth in order to test whether the HWC is a function of distance. The data associated with demography and socio-economy was aggregated and decoded for confidentiality of the respondents.

We conducted multivariate logistic regression with the entire independent variables [total of 11 variables: temporal (year, month, season, time), socioeconomic (gender, occupation and age group of the victims), and spatial (district, elevation, distance between point of attack and forest), and land use (forest, agricultural land, road and settlement)] in the model to understand the relationship between predictive and explanatory variable (human death and injury due to wildlife attack). However, no single variable independently predicted the attacks in the analysis, and this could be a reason of the large volume of dataset, and the diverse nature of variables.

We conducted two different statistical approaches following Naha et al. [52]. Chi-square test of independence was used to understand the association between temporal (year, month, season and time), and socio-demographic variables (gender, age and occupation) with the wildlife attacks. We classified victims into four age groups, < 20, 21–40, 41–60 and > 61 years. The association between socio-demographic and temporal variables and the attacks were analyzed using Pearson chi-square test (Table 1). For the spatial dataset, we conducted a generalized linear model with binomial distribution to predict the effect of variables on the wildlife attacks following Acharya et al. [6]. We used a priori candidate model and ranked them based on Akaike Information Criterion (AIC) values. Those models with lowest AIC values were

**Table 1. Association between socio-demographic, ecological variables and human wildlife conflict.**

| Variables and sample number | | Coefficients | | |
|---|---|---|---|---|
| | | Chi-square | df | p value |
| Gender | Female (D = 3, I = 13) | 0.594 | 1 | 0.440 |
| | Male (D = 5, I = 56) | | | |
| Age | 0–20 yrs (D = 6, I = 6) | 24.118 | 3 | 0.002 |
| | 21–40 yrs (D = 0, I = 19) | | | |
| | 41–60 yrs (D = 1, I = 32) | | | |
| | 61–84 yrs (D = 1, I = 12) | | | |
| Occupation | Farmers (D = 2, I = 46) | 19.587 | 2 | 0.005 |
| | Forest product collectors and passers by (D = 0, I = 15) | | | |
| | Others (D = 6, I = 8) | | | |

df = degree of freedom, D = death, I = injured.

considered the appropriate for explaining the wildlife attacks (Table 2). A model with a ΔAIC (the difference between the two AIC values being compared) of ≤ 2 is considered significantly better than the model it is being compared to. The number of attacks per year was summarized in terms of mean (M) and number of events with regard to different spatial, temporal and socio economic variables were analyzed by using percentage. The variability was recorded in terms of SE and at confidence interval of 95%. Statistical analyses were undertaken in R statistical software (R Core Team 2016; Version 1.0.44).

## Results

### Human wildlife conflict

Seven types of conflicts (crop raiding, livestock depredation, human attacks, traffic collision, disease transmission, property damage and mental distress) and associated 663 HWC cases were recorded in five study districts between 2011 and 2019, based on our review and field observations. Among the HWC types, livestock depredation and crop raiding were the most common. There were 77 (12%) cases pertaining to wildlife attacking to human, and these were used for further analyses (Fig 2). Out of these 77 cases, 67 (87%) cases were involved with Leopard and nine with Himalayan black bear.

**Table 2. Akaike Information Criterion (AIC) scores of generalized linear models with binomial structure predicting human attacks by wildlife in Parbat, Lamjung, Syangja, Tanahun and Kaski (PLSTK) districts, Nepal.**

| Models | df | AICc | ΔAIC |
|---|---|---|---|
| District | 5 | 49.2 | 0 |
| District + Distance from forest | 6 | 50.9 | 1.71 |
| District + Elevation | 6 | 51.2 | 2.02 |
| District + Land use | 8 | 51.4 | 2.21 |
| Elevation | 2 | 51.5 | 2.36 |
| Elevation + Land use | 5 | 52 | 2.83 |
| District + Elevation + Land use | 9 | 53.1 | 3.93 |
| District + Distance from forest + Elevation | 7 | 53.2 | 3.97 |
| Null | 1 | 53.4 | 4.24 |

ΔAIC is the difference between the AICc value of the best-supported model and successive models, and df = degree of freedom, Δ = delta.

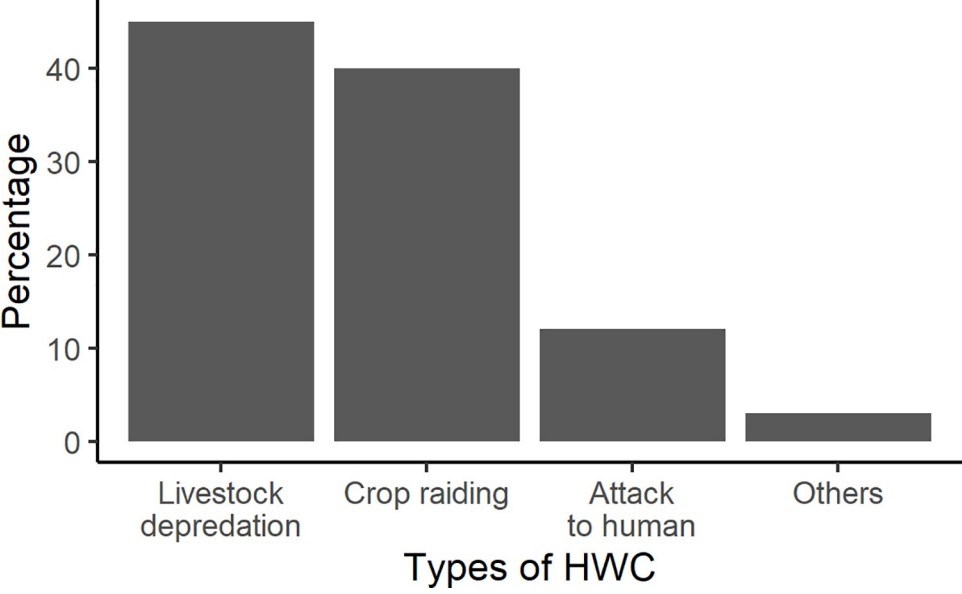

**Fig 2. Types of HWC in PLSTK districts during 2011 to 2019.**

## Temporal variations of attacks

The mean number of humans attacks per year was 8.55±1.15 (injuries 7.66 ± 1.13 and the deaths 0.88 ± 0.84). Incidence of human attacks was increasing over the period of nine years, ($\chi2 = 0.232$, df = 76, $p > 0.05$) (Fig 3). The frequency of human attacks varied with month and season ($\chi2 = 152.08$, df = 76, $p < 0.001$, and $\chi2 = 13.39$, df = 3, $p < 0.05$ respectively) with highest frequency of human attacks occurred in winter (December-March) and lowest in summer (July-August) (Fig 3). A high proportion of attacks (42%) occurred between September and December. The frequency of attacks varied with time of the day ($\chi2 = 134.45$, df = 76, $p < 0.001$). The highest percentage of attacks (45% injuries and 62% killings), was recorded between 15.00 pm and 19.59 pm.

## Socio-demographic characteristics of victims

Of the 77 victims, 79% (n = 61) were male and 21% (n = 16) were female. There was no difference in the proportion of male and female victims ($\chi2 = 0.59$, df = 1, $p > 0.05$). Victims' ages ranged from 3 to 84 years at the time of attack. Most victims (42%, n = 33) were 41 to 60 years old followed by 24% (n = 19) 21 to 40 years old. The chi-square test showed a significant difference in occupation and age groups of the victims ($\chi2 = 24.11$, df = 3, $p < 0.05$) (Table 1).

## Special pattern of human attacks

Between 2011 and 2019, eight people were killed and 69 were injured in five districts. All eight human deaths, six in Tanahun and two in Syangja were attacked by common leopard. Of the 77 human casualty cases, 6 human deaths and 16 injuries were reported from Tanahun followed by 18 injuries in Lamjung and 16 injuries and 2 deaths in Syangja. The lowest human injuries (n = 5) were recorded for Parbat district. Within the district, Bhanu municipality of Tanahun district was the most vulnerable to leopard attack and six children were killed within a period of two years between 2018 and 2019. Besides Leopard attack, Tanahun and Lamjung districts were also vulnerable to bear attacks (Table 2). Attacks by bear were mostly confined

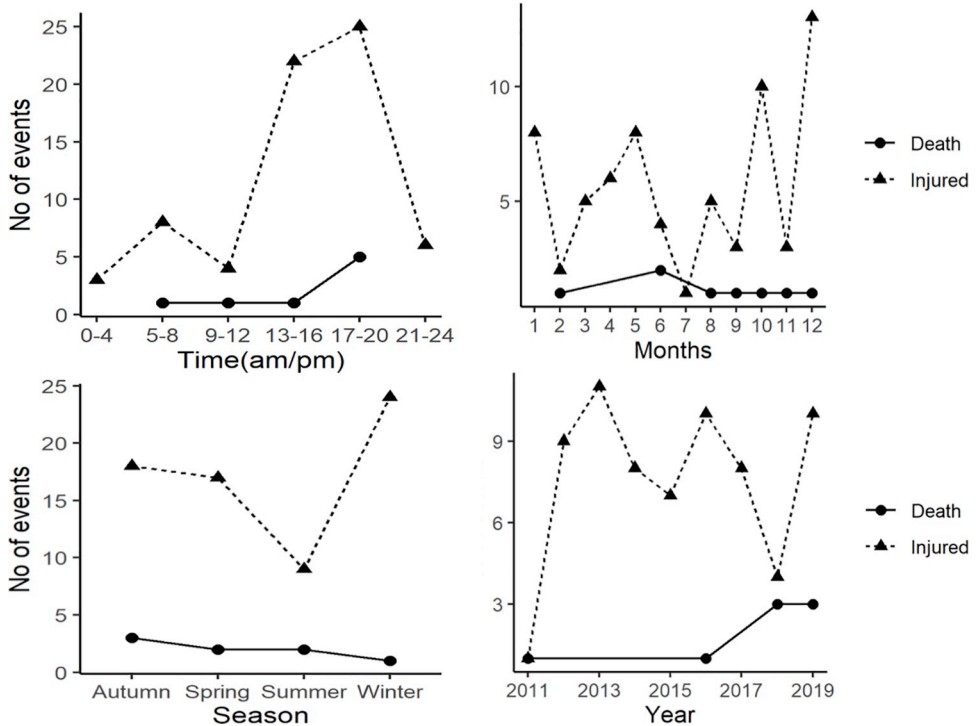

**Fig 3. Temporal distribution of HWCs in PLSTK districts during 2011 to 2019 showing time month, season and year of the events.**

inside forest. In contrary, the leopard attacks were recorded mostly from nearby of settlement and agriculture lands. Forty-eight percent (n = 37) attacks were reported at human settlement areas. There were 27% attacks in agriculture land (n = 21) and 24% (n = 19) in forest (Fig 4). GLM analysis on spatial variables demonstrated district location as top important variable for predicting wildlife conflict followed by distance from forest (Table 2). The probability of human attacks increases with decrease in elevation (β = -0.0021, Z = -1.762, *p* = 0.078), and distance from the forest (β = -0.608, Z = -0.789, *p* = 0.429).

## Discussion

### Human attacks

DFO records of five districts (Parbat, Lamjung, Syangja, Tanahun and Kaski) shows that mainly two wild animals (leopard and Himalayan black bear) involved in attacking humans while there are 26 wildlife species causing human-wildlife conflicts and 13 triggering severe human attacks in Nepal [53–55]. Of these animals, leopard attacked 67 people and Himalayan black bear injured nine indicated that the leopard attacks were not accidental and severely threatened local people and their livelihood. The studies in other mountain areas of the world also found that Himalayan black bear and leopard are the major mammals responsible for human attacks [39, 40, 55–58]. Big mammals like elephant, tiger and leopard are potentially dangerous in causing property destruction and inflicting injuries to people elsewhere [59]. There were 441 leopard attacks in Nepal over a decade between 1994 and 2004 [58]. Over two third leopard attacks were recorded from human settlement areas and agriculture land. This shows that the human leopard conflict has been a serious issue in Nepal for a long time.

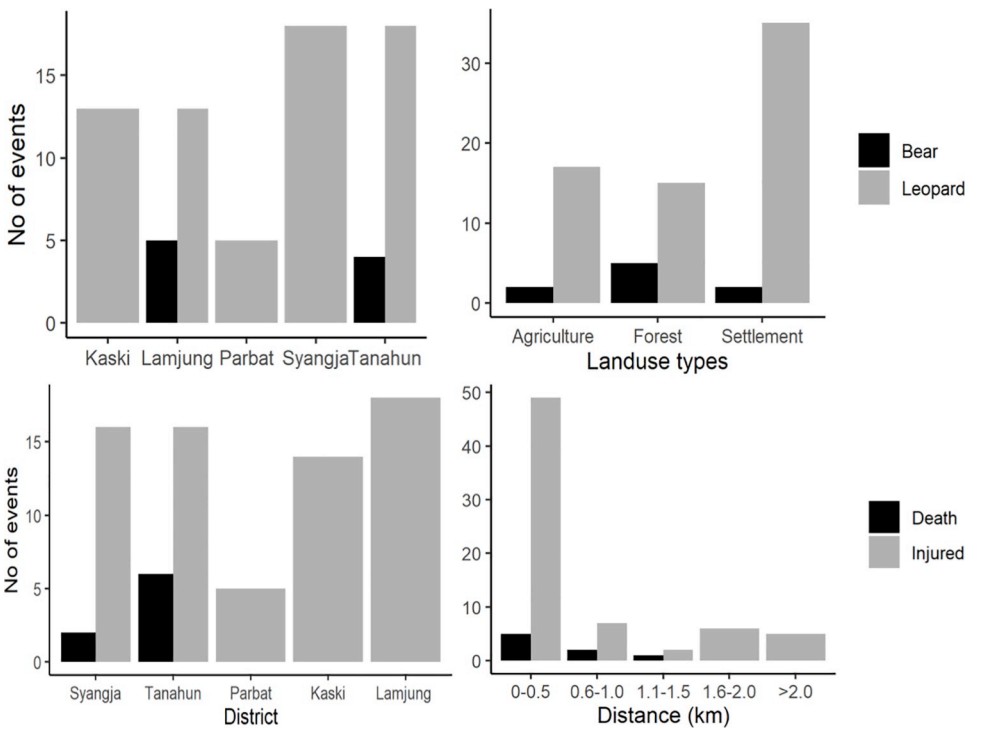

**Fig 4. Spatial distribution of HWCs in PLSTK districts during 2011 to 2019 showing events by districts, land use type and distance of the event site from the forest.**

The average number of attacks per year was 8.55 (SE 1.15), including 7.66 (SE 1.13) injuries and 0.88 (SE 0.84) deaths. Similar finding of average seven injuries in a year was recorded in Chitwan district, central Nepal [23]. The records in our study area were less than that of Pauri Garhwal, India where the average injuries and death per year were 11 (SE 1.13) and 3 (SE 0.6) respectively [60]. The higher number of attacks in Pauri Garhwal is probably due to the higher population density (130/sq km) than our study area (110/Km$^2$). The Pauri district (area 5230/ Km$^2$) of India bordering the western Nepal has similar mid-hill types of physiography and sub-tropical climate. Similarly, the forest cover in Pauri is higher (64%) than our study area (32.5%), which may inhabit larger population of leopard.

Human attacks by Himalayan black bear are considered to be the most ubiquitous form of conflict in the mid-hills of Nepal [55]. All nine attacks by Himalayan black bear were recorded at Tanahun and Lamjung districts. Similarly, within the Panchase Protected Forest, which lies across three study districts (Kaski, Syangja and Parbat) four cases of human casualty were associated with Himalayan black bear [26]. Panchase is the area where the agro-pastoralism is still prevalent. Human casualty is common in mid-hills where the livelihood is agro-pastoralism. A study of Nepal Trust for Nature Conservation (NTNC) from 2005 to 2013 [57] catalogued six cases of Himalayan black bear attacks in Manaslu Conservation Area, Gorkha, Central Nepal. It indicates that the Himalayan black is also a major wildlife causing HWC in mid-hill physiographic region of Nepal.

## Distribution of HWC

Our study area is among the areas of highly vulnerable to HWC. Despite the vulnerability, the human attacks in our five study districts were varied with magnitude. There were eight death

reports, all from the leopard attacks and only occurred in two districts (six in Tanahun and two in Syangja). It might be due to the higher habitat destruction and less availability of prey species in forest in these districts. In Nepal, 75 out of 77 districts have been suffering from HWC, with Jhapa, Baitadi, Tanahun, Kaski, Arghakhanchi, Manang, Mustang districts being the most suffered [58]. Tanahun and Syangja districts truly represent the mid-hill physiography are found to be the most vulnerable for HWC. Tanahun district has the highest livestock holding size (3.72) compared to the rest districts [61]. There were 12 people killed by leopard in Kaski district between 1987 and 1989 [62] indicating the prevalent and persistent of HWC in mid-hills. Leopard is common in mid-hills [63]. Thapa [64] reported that 45 humans were attacked and 14 were killed by leopard in Chitwan Annapurna Landscape (CHAL) between 2006 and 2013. There were seven human injuries by leopard in 2009 in Chitwan district, central Nepal [23]. There were 20 confrontations of leopard in Kathmandu valley between 2010 and 2013 [37]. All these incidences revealed that the central Nepal and mid-hill districts are highly vulnerable to leopard attack [18].

The forest habitats of the mid-hills are surrounded by the large human population. Dense population along with local poverty of people trigger the human interference in wildlife habitat [65] and aggravate HWC at local scale. The hills and mountains have highest percentage of people under poverty (42.3%) and high emigration rate on average, and these have increasing trends [45]. Outmigration caused the lack of working manpower in agriculture land and farmland resulting in abandonment of agriculture activities. Fallow land promotes the growth of shrubs and bushes connecting wildlife habitat with human settlements. These, in combination of prey shortage in the natural habitats, have facilitated the HWCs [45].

Forty eight percent of the wildlife attacks (n = 37) reported were from human settlement areas followed by that from agriculture land 27% (n = 21) and forest 24% (n = 19), revealing that the most cases happened in anthropogenic landscapes. Generally, the probabilities of attacking by large carnivores are reported to increase with dense forest [66] but our results suggest that higher risk of human attacks by wildlife are in human settlements. Out of total 77 incidents, leopard attack sites were all in human settlement area and agriculture lands and 75% (n = 58) of these were found within one km distance from the nearest forest. Geographical features of the mid-hills with undulating landscape, presence of small creeks, trees and bushes within and nearby village may make favorable condition for leopard to interact with people. Furthermore, leopard is a generalist species and adapted to living in the forest fringes, nearby human habitations and moderate vegetation cover [60, 67] and avoid confronting with bigger carnivore mammals such as tigers [68]. Leopard tolerate proximity to humans better than bears, lions and tigers and often come into conflict with humans [69]. Tiger is frequent at lowland Tarai region below 1000 m in Chitwan National Park (100 m– 815 m) and Bardia National Park (150 m– 1400 m) [70] and hardly seen in middle mountains [71]. Increasing HWC in mid-hills is likely due to higher dependency of people on forest and the area being extensively extended outside protected area system. Due to the wider spread poverty in study area, people often go to forest for collecting forest-based fruits, vegetables, nuts etc. [33] and poaching the wildlife to sustain their livelihood [54]. Studies showed that poaching of prey species and habitat degradations are rampant resulting in high number of HWCs in Nepal [6, 18, 64].

## Temporal pattern of HWC

The frequency of human attacks varied with time, month, season and year. There were 52% attacks between 15.00 pm and 19.59 pm in study area, which contrasts with the pattern of leopard attack in Pauri Garhwal where the most attacks (53%) were reported during day time (08.00 am –16.00 pm) [52]. The leopard also attacks in early and late night [72]. We recorded

31% leopard attacks (n = 24) between 17.00 pm and 19.59 pm supporting the crepuscular nature of leopard. Fifty-six percent of human attacks happened in autumn and winter season. This was associated with the frequent movement of human for crop harvesting and forest product collection during this period. The increased human activities was reported as a primary reason of rising leopard attack in India also [73].

The second largest number of human attacks was associated with Himalayan black bear and the most of them were reported between October and February. Our result substantiated the earlier results [35, 74] which reported that the early winter is likely to have greater number of attacks by Himalayan black bear in the mid-hills. Attacks by Black bears during late August to September were recorded high in the Sichuan Province in China coincide with wild mushroom harvesting [75]. In India majority of the crop depredation by Himalayan black bear and conflict with people occurred in between August and September [76] which coincided with the time of visitation by rural people for fodder, grasses and firewood [77]. Black bears hibernates at the end of autumn and on the pre-hibernation period they become more active, consume more grains and fruits and travel long distance in search of their preferred food [78], which may have resulted in higher confrontation of black bear with people.

## Characteristics of victims

The middle-aged people (41–60 years old) were attacked more because they were more likely to engage in outdoor occupations. Generally male are engaged in outdoors, and they were found to be frequently attacked (61, compared to 16 of females) by wild animals. Farmers were the major victims of animal attack ($p < 0.05$). As leopard frequently range and refuge near human settlements, farmers in their settlement and farmland are often victimized [69]. Forest product collectors who venture several kilometers into the forests are also likely to encounter with wild animals.

## Causes and impacts

Human attacks by wildlife were found increasing in recent years. There is an increasing trend in the number of human attacks due to wildlife interactions, even in the areas with no previously reported incidents [13, 14]. This trend was consistent to increasing forest cover and PAs management [26]. We reported an increasing forest cover (388 ha/yr) in the PLSTK districts between 1996 and 2016 [69]. Increasing forest cover was also accounted between 2000 and 2010 in CHAL districts [79, 80]. There were several reports of increasing HWC as increasing forest cover [81], attributed by the successful community-based forest conservation programs in the mid-hills and free movement of wild animals to nearby agricultural lands [28, 82–84].

On the other hand, deforestation and forest degradation have led to frequent HWC [85]. In the mid-hills of Nepal, animal husbandry, forest product collection and agro-pastoralism create competition between local communities and wildlife for the use of natural resources. The situation was worsened as the land-use change resulted population decline of prey species, resulting in increased incidence of wild animal preying on livestock and escalated HWC [85]. Nowadays, because of the increasing urbanization and land use change [86], land abandonment and unattendance of agricultural lands transformed areas into shrubland covered by invasive plant species [87]. These regenerating areas enabled wild animals to approach human settlement and result in HWC. Increasing HWC due to both increasing and decreasing forest cover is a conservation paradox. The majority of HWC occurred in nearby human-dominated landscapes highlights the need for proper management of areas outside PAs. Thus community-based forest conservation to yield sustainable supply of forest products and improve prey population through wise use of lands is crucial in HWC management.

## Human-wildlife conflict management

There are some strategies proposed and implemented to mitigate HWC in Nepal [88].

Control of habitat fragmentation and degradation of wildlife is the foremost one. Carrying out habitat management activities within forest to provide the necessary food, water and shelter to wildlife within forest area is also crucial to decrease the invasion of wildlife in villages. Decreasing dependency of people on forest resources by providing alternate livelihood options such as bee keeping, horticulture of citrus species, and cultivation of spices like turmeric and ginger etc. can help reduce HWC and improve the livelihood of rural communities. Compensation mechanism is a major strategy to promote the human wildlife co-existence. Government of Nepal has started to distribute the compensation / relief fund to the wildlife victimized people for their loss of property and life since 2012. However, its procedures are arduous and time consuming. Thus, a simple procedure of compensation and establishing it at local level will help more effectively mitigate the HWC impact. Removing bushes and invasive species around human settlements should be initiated to reduce wildlife attacks. Without taking into account of spatio-temporal variability and the fate of wildlife seem to be ineffective in mitigating conflicts on a long-term basis [56].

## Conclusions

Based on the findings of present study, we conclude that patterns of wildlife attacks on humans are influenced by spatial and temporal factors. Leopard was the major wildlife cause HWC followed by black bear. The conflict ranged from crop raiding, livestock depredation, traffic collision, property damage, transmission of diseases and human attacks. Among them, human attack was the most critical expression of HWC and needs addressing sensitively to liaison the local support for wildlife conservation.

The wildlife attack to human has been increasing and majority of attacks occurred in and around human-dominated landscape, followed by forest. Control of habitat fragmentation, degradation and implementation of habitat management activities within forest are crucial to decrease the invasion of wildlife in human settlement area. Decreasing dependency of poor people on forest by providing alternative livelihood opportunity would helpful to decrease the encounter of wildlife with people. Moreover, simplifying the compensation system to the people victimized by wildlife is necessary to promote human wildlife co-existence.

## Supporting information

**S1 File. Data and associated variables used for this study.**
(CSV)

## Acknowledgments

We thank study area residents and DFOs who responded to our queries and survey. S Bhandari, B Adhikari, and anonymous reviewers provided valuable comments on earlier drafts of this manuscript.

## Author Contributions

**Conceptualization:** Kedar Baral, Achyut Aryal, Weihong Ji.

**Data curation:** Kedar Baral, Hari P. Sharma, Bhagawat Rimal, Khum Thapa-Magar.

**Formal analysis:** Kedar Baral, Hari P. Sharma, Bhagawat Rimal, Khum Thapa-Magar, Ripu M. Kunwar, Achyut Aryal, Weihong Ji.

**Methodology:** Kedar Baral, Hari P. Sharma.

**Resources:** Rameshwar Bhattarai.

**Supervision:** Ripu M. Kunwar, Achyut Aryal, Weihong Ji.

**Visualization:** Rameshwar Bhattarai, Ripu M. Kunwar.

**Writing – original draft:** Kedar Baral.

**Writing – review & editing:** Kedar Baral, Bhagawat Rimal, Khum Thapa-Magar, Rameshwar Bhattarai, Ripu M. Kunwar, Achyut Aryal, Weihong Ji.

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
