## [Decision Letter · Decision Letter 0]

15 Jul 2021

PONE-D-21-16754

Characterization and management of human-wildlife conflicts in mid-hills outside protected areas of Gandaki province, Nepal

PLOS ONE

Dear Dr. Kunwar,

Thank you for submitting your manuscript to PLOS ONE. After careful consideration, we feel that it has merit but does not fully meet PLOS ONE’s publication criteria as it currently stands. Therefore, we invite you to submit a revised version of the manuscript that addresses the points raised during the review process.

The authors must address all the comments of the reviewers specially reviewer #1 in the revised manuscript.  

We look forward to receiving your revised manuscript.

Kind regards,

Lalit Kumar Sharma

Academic Editor

PLOS ONE

5. We note that Figure 1 in your submission contain map images which may be copyrighted. All PLOS content is published under the Creative Commons Attribution License (CC BY 4.0), which means that the manuscript, images, and Supporting Information files will be freely available online, and any third party is permitted to access, download, copy, distribute, and use these materials in any way, even commercially, with proper attribution. For these reasons, we cannot publish previously copyrighted maps or satellite images created using proprietary data, such as Google software (Google Maps, Street View, and Earth). For more information, see our copyright guidelines: http://journals.plos.org/plosone/s/licenses-and-copyright.

Reviewers' comments:

Reviewer's Responses to Questions

**Comments to the Author**

1. Is the manuscript technically sound, and do the data support the conclusions?

Reviewer #1: Partly

Reviewer #2: Partly

2. Has the statistical analysis been performed appropriately and rigorously? 

Reviewer #1: Yes

Reviewer #2: Yes

3. Have the authors made all data underlying the findings in their manuscript fully available?

Reviewer #1: No

Reviewer #2: Yes

4. Is the manuscript presented in an intelligible fashion and written in standard English?

Reviewer #1: No

Reviewer #2: No

5. Review Comments to the Author

Reviewer #1: This paper deals with human attack by wildlife, but the scope of the title is quite bigger. All statistical tests are done rigorously, but the result section is not showing many important details such as coefficients of explanatory variables. Inclusion of visual representation of the model results would make this paper more useful to the managers. Detailed comments are given in the manuscript.

Reviewer #2: The study does show a pattern of Human wildlife interactions outside Protected Areas in some districts of Nepal, the information is based on secondary data and no effort has been made for the primary data collection. However, this secondary data information is also worth, if presented in a crisp and organized manner. this manuscript needs a complete revision interms of providing the facts and supporting with the similar references. the information provided in the manuscript is chaotic and needs re organisation. As noted in previous review also, the ethinicity as a variable needs to be spelled out clearly with a proper justification w.r.t Human wildlife interactions.

6. PLOS authors have the option to publish the peer review history of their article (what does this mean?). If published, this will include your full peer review and any attached files.

Reviewer #1: **Yes: **Tapajit Bhattacharya

Reviewer #2: **Yes: **Dr Samina Amin Charoo

---

## [Author Response · Author response to Decision Letter 0]

5 Oct 2021

This paper was edited by Professor Weihong Ji

Head of Human-Wildlife Research Group

Postgrad subject leader for Zoology, Ecology and Conservation Biology

School of Natural and Computational Sciences

Massey University Albany

Private Bag 102 904 North Shore Mail Centre Auckland, New Zealand

---

## [Editor Report · Decision Letter 1]

12 Oct 2021

PONE-D-21-16754R1Characterization and management of human-wildlife conflicts in mid-hills outside protected areas of Gandaki province, NepalPLOS ONE

Dear Dr. Kunwar,

Thank you for submitting your manuscript to PLOS ONE. After careful consideration, we feel that it has merit but does not fully meet PLOS ONE’s publication criteria as it currently stands. Therefore, we invite you to submit a revised version of the manuscript that addresses the points raised during the review process.

We look forward to receiving your revised manuscript.

Kind regards,

Lalit Kumar Sharma

Academic Editor

PLOS ONE

Additional Editor Comments:

I could see the authors have attempted addressing the comments of both the reviewers. However, it is becoming difficult to track changes in the revised version because of a lot text has been changes after the comments. Moreover, the response to reviewers is not properly provided. Hence, it is imperative that the authors must adhere to the point to point policy. It will be good if the author provide all explanations point to point in the response to reviewers file. One of the reviewer has provided comments in the manuscript which needs to be explained in the response to reviewer file along with the line and page number so the revisions can be tracked.
---

## [Author Response · Author response to Decision Letter 1]

22 Oct 2021

Dear Editor, 

Dr. LN Sharma

PLOS One Journal

We are thankful to you and reviewers for providing us the constructive comments. We truly appreciated your cooperation in this regard. 

Please find our revised MS language edited by Prof. Dr. Weihong Ji, Massey University New Zealand, and Dr. Craig Morley, TO Institute of Technology, New Zealand. 

We have uploaded a clear MS “Manuscript”, a marked copy of the MS “Revised Manuscript” and 3 “Response to Reviewers’ Comment” Sheets. We received 2 reviews on July 15th, and one review on Aug 28th. We also received Edit Requested on Oct 6th and Revision Required on Oct 13th. 

We tried our best to address the comments, locate the corrections along with the provision of line numbers, changed texts and associated explanation. We prepared the Reviewers Comment address sheets by point-by-Point method. As we have received 4 reviews on July 15, Aug 28, Oct 6 and Oct 13, we addressed them all and prepared the Comment Address sheet separately for ease of your referencing. Please find all corrections in the marked copy. 

We truly appreciated your cooperation in this regard. Thank you again and please let me know if you need any other supplementary information regarding this MS and revisions. 

Sincerely,

Ripu Kunwar

ripukunwar@gmail.com

rkunwar@fau.edu

---

## [Editor Report · Decision Letter 2]

8 Nov 2021

Characterization and management of human-wildlife conflicts in mid-hills outside protected areas of Gandaki province, Nepal

PONE-D-21-16754R2

Dear Dr. Kunwar,

We’re pleased to inform you that your manuscript has been judged scientifically suitable for publication and will be formally accepted for publication once it meets all outstanding technical requirements.

Kind regards,

Lalit Kumar Sharma

Academic Editor

PLOS ONE
---

## [Editor Report · Acceptance letter]

11 Nov 2021

PONE-D-21-16754R2 

Characterization and management of human-wildlife conflicts in mid-hills outside protected areas of Gandaki province, Nepal 

Dear Dr. Kunwar:

I'm pleased to inform you that your manuscript has been deemed suitable for publication in PLOS ONE. Congratulations! Your manuscript is now with our production department. 

Kind regards, 

on behalf of

Dr. Lalit Kumar Sharma 

Academic Editor

PLOS ONE